# OpenReview forum: "Breaking the Code: Security Assessment of AI Code Agents Through Systematic Jailbreaking Attacks"
_ICLR.cc/2026/Conference — ICLR 2026 Conference Desk Rejected Submission_

### Official Review · Reviewer_XHKK · 2025-10-26

**Soundness:** 2
**Presentation:** 3
**Contribution:** 2
**Rating:** 2
**Confidence:** 5

**Summary:**

This paper presents CodeAgentJail, a benchmark covering three workspace regimes: empty (CAJ-0), single-file (CAJ-1), and multi-file (CAJ-M). It is paired with a hierarchical, executable-aware Judge Framework that evaluates (i) compliance, (ii) attack success, (iii) syntactic correctness, and (iv) runtime executability. The authors conduct experiments on different models built upon OpenHands and find that agent frameworks tend to be more vulnerable than base models.

**Strengths:**

1. The paper is clearly written and easy to follow.
2. The multi-file setting is interesting and realistic.

**Weaknesses:**

W1. Limited novelty in benchmark construction design

The benchmark contribution seems limited. Levels 0 and 1 are largely inherited from RMCBench, while level 2/M is generated by prompting an uncensored model to produce detailed code. However, it is unclear whether the generated code truly aligns with malicious purposes and is both practical and executable as intended.

W2. Overlap with prior executable-aware benchmarks

The paper claims:

> “Prior evaluations emphasize refusal or harmful-text detection, leaving open whether agents actually compile and run malicious programs. To our knowledge, this is the first executable-aware benchmark for code agents across all three workspace settings.”

However, previous work (e.g., RedCode) also emphasizes executable-aware evaluation of risky and malicious program generation and execution.

W3. Reliability of LLM-based executability judges

The proposed executability judge is itself an LLM-based agent. This raises concerns about consistency and reliability—e.g., a “attack evaluation judge” may consider a sample malicious, while the “executability judge” might not.

Moreover, the claim that the judge “never edits or overwrites existing files” is unconvincing, as the agent may still inadvertently modify the workspace. Additionally, agents could refuse to execute potentially malicious code, which complicates runtime evaluation.

**Questions:**

Q1: What are the specific design innovations in the benchmark construction compared to prior work?

Q2: What additional significant contributions does this work provide beyond RedCode [1]?

Q3: How do the authors ensure that the LLM-based runtime evaluation judge remains reliable, especially given the malicious or unsafe nature of the evaluated tasks?


[1] RedCode: Risky Code Execution and Generation Benchmark for Code Agents (NeurIPS 2024)

---

> ### Author Response · Authors · 2025-11-20
> **Author Reply (Part 1)**
>
> We appreciate the reviewer’s feedback and effort. Thanks to the reviewer for finding our benchmark both realistic and interesting.
>
> > “The benchmark contribution seems limited. Levels 0 and 1 are largely inherited from RMCBench, while level 2/M is generated by prompting an uncensored model to produce detailed code. However, it is unclear whether the generated code truly aligns with malicious purposes and is both practical and executable as intended.”
>
> Thank you for this thoughtful comment. We agree that CAJ-0/1 reuses prompts from RMCBench, and this is **intentional**: it allows us to anchor our results to an existing, well-studied malicious code benchmark while shifting the setting from base LLMs to autonomous code agents. The main benchmark contribution of CAJ-Bench is not simply reusing prompts, but:
>
> * **Re-structuring the tasks into agent-centric regimes** (empty → single-file → multi-file workspaces) that mirror escalating attacker capability and code context, and
> * **Introducing the new CAJ-M split**, where we construct multi-file malicious repositories explicitly designed for end-to-end execution and evaluation within code agents (with imports, entrypoints, and run scaffolds).
>
> On the quality and malicious alignment of CAJ-M, we agree that this needs to be substantiated. In response to this concern, we have:
>
> * Used carefully designed prompts to an uncensored model that specify the malware category, intended behavior, and required multi-file structure (including an executable entrypoint and minimal run instructions).
> * For this rebuttal, additionally applied **VirusTotal’s Code Insight API and CodeQL analysis, together with manual inspection**, to verify that the resulting repositories implement behavior consistent with their malicious labels and are practically executable.
>
> We will add a dedicated description of this CAJ-M construction and validation procedure to the paper.
>
> Finally, regarding practicality and executability: our runtime-error-free rates on CAJ-M provide **empirical evidence that the tasks are executable as intended**. For the stronger models (e.g., GPT-4.1, GPT-o1, DeepSeek-R1, Qwen3), a substantial fraction of CAJ-M instances compile and run successfully in our sandboxed environment. If the underlying repositories were not realistic or executable, we would not observe such high end-to-end success rates.
>
> > “*“Prior evaluations emphasize refusal or harmful-text detection, leaving open whether agents actually compile and run malicious programs. To our knowledge, this is the first executable-aware benchmark for code agents across all three workspace settings.”*
> > However, previous work (e.g., RedCode) also emphasizes executable-aware evaluation of risky and malicious program generation and execution.”
>
> Thank you for pointing this out and for highlighting the RedCode work. You are absolutely right that RedCode already conducts executable-aware evaluation of risky and malicious program generation and execution for code agents. We apologize for not positioning our work relative to RedCode more clearly in the initial submission. In the revised version, we will explicitly position our contribution as **complementary to RedCode rather than prior to it**. In particular:
>
> * RedCode conducts executable-aware safety evaluation of code agents via RedCode-Exec (risky code execution across diverse vulnerability scenarios) and RedCode-Gen (harmful software generation across multiple malware families in sandboxed environments)
> * CAJ-Bench is designed as a systematic, multi-level jailbreaking benchmark that **explicitly varies the workspace setting and attacker capability** (CAJ-0: empty, CAJ-1: single-file partial, CAJ-M: multi-file malicious repositories). Our goal is to study **how added malicious code context affects** (i) refusal/compliance, and (ii) the fraction of attacks that become fully executable across these three regimes.
>
>
> So, while RedCode and our work both care about executability, CAJ-Bench contributes a structured, three-level workspace design and a multi-judge pipeline tailored to jailbreaking scenarios, which we see as a **natural next step, building on ideas like RedCode**.
> We will update the paper to (1) properly cite RedCode as earlier executable-aware work on code agents, (2) tone down any “first” wording, and (3) clearly articulate how CAJ-Bench extends this line of research by introducing a multi-level, attacker-capability-aware framework for jailbreaking code agents.

---

> ### Author Response · Authors · 2025-11-20
> **Author Reply (Part 2)**
>
> > “The proposed executability judge is itself an LLM-based agent. This raises concerns about consistency and reliability—e.g., a “attack evaluation judge” may consider a sample malicious, while the “executability judge” might not.”
>
> Great question, and we appreciate the chance to clarify this.
>
> First, our Attack Evaluation Judge and our Executability Judges are **designed to answer different questions and do not override each other**:
>
> * Attack Evaluation Judge (LLM-based): “Is this code malicious in intent/function?”
> * Syntax-Error Judge (tool-based micro-agent): “Does this malicious code parse/compile across all relevant files?”
> * Runtime-Error Judge (tool-based micro-agent): “Does this malicious code actually run end-to-end in our sandbox (exit code 0, no crashes)?”
>
>
> Only samples that the Attack Evaluation Judge has already labeled as harmful are passed on to the syntax and runtime stages. The executability judges **never re-label something as non-malicious**; they only determine whether that malicious artifact is deployable (syntax- and runtime-correct) or not.
>
> So, in the scenario you raised: “Attack Evaluation Judge considers a sample malicious, while the executability judge might not.”; this is exactly the behavior we want and one of the main motivations for our pipeline:
>
> * Such a sample is still **counted** as harmful for ASR (because its logic/intent is malicious),
> * but it is **excluded** from “+ Syntax-Error-Free Rate” and “+ Runtime-Error-Free Rate” because it fails to parse or run.
>
>
> This lets us explicitly distinguish:
> 1. **Seemingly harmful but undeployable code** (malicious intent, broken in practice), from
> 2. **Fully deployable attacks** (malicious + syntactically valid + runtime-correct).
>
>
> When we report **“+ Runtime-Error-Free Rate”**, it is by construction the fraction of prompts for which:
>  (i) the agent complied,
>  (ii) the artifacts are judged harmful, and
>  (iii) those harmful artifacts parse and execute without errors in our sandbox.
>
> Codes that the Attack Evaluation Judge marks as harmful but that fail syntax or runtime checks do not contribute to that final metric; they remain harmful, but not executable.
>
> We will revise the paper to make this hierarchy explicit and to clarify that “inconsistencies” of the form you describe are **not a bug but a feature of the design: they reveal precisely the gap between harmful intent and deployable harm** that prior refusal-only or text-only evaluations cannot capture.
>
> > “Moreover, the claim that the judge “never edits or overwrites existing files” is unconvincing, as the agent may still inadvertently modify the workspace. Additionally, agents could refuse to execute potentially malicious code, which complicates runtime evaluation.”
>
> > “How do the authors ensure that the LLM-based runtime evaluation judge remains reliable, especially given the malicious or unsafe nature of the evaluated tasks?”
>
> Thanks for raising this concern. First, regarding modifications: our runtime executability judge is configured as a **non-mutating micro-agent**. Unlike the main code agent, it is **not given any file-editing or write access** – only read/inspection and run capabilities inside the container. Second, our judge is based on the OpenHands agent framework (June version), which does not have any safeguard feature that can detect malicious code at runtime and refuse to execute it. For safety, we instead rely on **sandbox isolation** during execution, so any malicious behavior is contained while still allowing us to faithfully assess executability.

---

> > ### Author Response · Authors · 2025-11-20
> > **Author Reply (Part 3)**
> >
> > > “What are the specific design innovations in the benchmark construction compared to prior work?”
> >
> > Thank you for this question – we are happy to clarify the concrete design innovations. In terms of benchmark construction, CAJ-Bench introduces three key design choices compared to prior work:
> >
> > 1. **Attacker-capability–aware workspace regimes (CAJ-0/1/M).**
> > We explicitly structure the benchmark into three levels – empty workspace, single-file partial code, and multi-file malicious repositories – to mirror escalating attacker capability and code context. This lets us probe whether a code agent can (i) recognize and refuse maliciousness in isolation, (ii) when it is embedded in a single file, and (iii) when it is distributed across multiple files.
> >
> > 2. **Single-file and multi-file malicious repositories tailored for agents.**
> > Beyond reusing RMCBench prompts, we construct CAJ-1 hole-filling tasks and CAJ-M multi-file repos so that each instance is directly runnable by an autonomous code agent (with imports, entrypoints, and workspace structure).
> >
> > 3. **Integrated quality validation for maliciousness and practicality.**
> > To ensure that the benchmark truly reflects malicious and executable scenarios, for this rebuttal, we combine human evaluation, VirusTotal’s Code Insight API, and CodeQL analysis to validate that the codebases align with their malicious categories and are practically meaningful.
> >
> >
> > We believe these design choices make CAJ-Bench a useful tool for understanding where current code agents stand in terms of safety guardrails under varying attacker capabilities, and for providing a solid foundation on which future, more sophisticated attacks and defenses can be built.
> >
> > > “What additional significant contributions does this work provide beyond RedCode [1]?”
> >
> > Thank you for this question. As we mentioned earlier, we see our work as **complementary to RedCode rather than a replacement**.
> > RedCode benchmarks the safety of AI code agents along two axes: risky code execution (RedCode-Exec) and malware-style function generation (RedCode-Gen). In contrast, our work contributes **several additional, orthogonal pieces:**
> >
> > **Systematic, multi-level jailbreaking benchmark (CAJ-0/1/M).**
> > We explicitly structure CAJ-Bench into three workspace regimes – empty directory, single-file partial program, and multi-file malicious repository – to mirror escalating attacker capability (naive → capable → expert). This lets us study how *increasing malicious code context systematically erodes refusal and increases both ASR and executability*, especially in the single- and multi-file settings.
> >
> >
> > **Malicious programs embedded in realistic workspaces.**
> > Whereas RedCode-Gen focuses on malware-style functions, CAJ-Bench emphasizes full programs and repositories that an agent must navigate, modify, and complete. This design directly probes whether agents understand and react to maliciousness *when it is distributed across files and embedded in a realistic project layout*.
> >
> > **Hierarchical, executability-aware evaluation pipeline.**
> > We introduce a four-stage pipeline that separately tracks:
> >  (i) compliance vs. refusal,
> >  (ii) malicious intent/function (Attack Evaluation Judge),
> >  (iii) syntax/parse correctness, and
> >  (iv) runtime success in a sandbox.
> > *This exposes the gaps between “policy violation” and “deployable attack code”*, and quantifies how many jailbreaks actually result in runnable, end-to-end attacks across the three workspace regimes.
> >
> > We will revise the manuscript to make this relationship to RedCode explicit, framing our work as an extension of executable-aware evaluation into a multi-regime, attacker-capability-aware jailbreaking benchmark with a decomposed notion of deployable harm.

---

> > > ### Author Response · Authors · 2025-11-21
> > > **Paper Revision + Author Reply**
> > >
> > > Dear Reviewer,
> > >
> > > We are glad to let you know that we have carefully incorporated your suggestions into the revised version of the paper. In the uploaded revised PDF (edits marked in blue), the main changes addressing your comments are:
> > > * **CAJ-M quality validation:** We added validation of CAJ-M using VirusTotal Code Insight and CodeQL, together with manual inspection, in the last paragraph of Section 3 (CodeAgentJail Benchmark), to further substantiate that CAJ-M repositories implement their intended malicious behaviors.
> > > * **Positioning with RedCode:** We now (1) properly cite RedCode as earlier executable-aware work on code agents, (2) tone down any “first” wording, and (3) clearly articulate how CAJ-Bench extends this line of research by introducing a multi-level, attacker-capability–aware framework for jailbreaking code agents. These changes appear in the Introduction, Related Work, and Section 7.
> > > * **Judge pipeline clarification:** We introduced a new subsection, Section 4.3 (“Judge Framework”), to clearly describe how the four-stage judge pipeline (refusal → attack evaluation → syntax → runtime) is composed and to emphasize that syntax and runtime judges are only applied to artifacts already labeled harmful.
> > >
> > > In addition, we made the following improvements that are closely related to your concerns:
> > > * **Agent-agnostic ablation:** We added an ablation study extending CAJ-Bench to other code agents (SWE-Agent and a Codex-style agent from OpenAI) on CAJ-0, and report the results in Section 7 under “Extending to Different Code Agents.”, showing that our benchmark is agent-agnostic.
> > > * **Human alignment of robustness judges:** We performed human verification of the Refusal and Attack Evaluation Judges and added the results as a dedicated paragraph in Section 4.1.
> > >
> > > We hope these revisions address your concerns. Thank you again for your detailed and constructive feedback, which we believe has significantly improved the paper. Please let us know if you have any further questions or concerns.

---

### Official Review · Reviewer_z1ru · 2025-10-29

**Soundness:** 3
**Presentation:** 2
**Contribution:** 2
**Rating:** 4
**Confidence:** 3

**Summary:**

The paper proposes a benchmark to evaluate if a coding agent actually compile and run malicious programs generated by itself in response to jailbreaking queries. The benchmark covers three regimes - starting with an empty repo, completing a single-file codebase, and working in a multi-file codebase. The paper also proposes a judge framework testing compliance with the harmful request and attack success, along with syntactic correctness and executability. 7 LLMs are benchmarked, and show significant potential for misuse. As more context is provided in the codebase, the attack success rate increases. LLMs are shown to be more vulnerable than their base forms when they are wrapped in the OpenHands agentic framework.

**Strengths:**

- A three-tiered benchmark is provided for evaluating the jailbreak susceptibility of coding agents, covering different stages of completion of the codebase.
- An intuitive framework is provided for judging the success rates of the proposed attacks, with a 4-stage evaluation.
- Some interesting analysis is provided to explain why LLMs are more vulnerable to jailbreak attacks when wrapped in the OpenHands agentic framework, involving the overturning of refusal in latter stages of planning.
- Detailed analysis is provided to determine the most common classes of vulnerabilities present in contemporary coding agents.

**Weaknesses:**

- 2 splits of the codebase is derived from existing work, and no analysis is provided for the quality of the CAJ-M split, which was created by this work. Therefore, the proposed benchmark largely builds on existing work. How does CAJ compare with other coding jailbreak benchmarks in terms of scale?
- No analysis is provided to show whether the refusal judge and attack evaluation judge results in human-aligned evaluation of ASRs.
- While this work focuses on executability of generated code, it does not consider how easily the generated code can be debugged via an agentic framework to be made executable, as long as it encodes the malicious intent. This somewhat limits the practicality of this benchmark, in my opinion.
- While the paper claims that LLMs are more vulnerable when wrapped in an agentic framework, the only framework used in the paper is the OpenHands framework. An agentic framework with explicit checks for malicious intent would be more resilient to attacks than the base LLM.

**Questions:**

- Can this benchmark be used to evaluate Claude Code/Codex-style agentic frameworks?
- Does the executability judge approach introduced in this work generalize to other coding jailbreak benchmarks?
- In CAJ-1/CAJ-M, are checks done to ensure that the part of the file that the agent is tasked with filling in also involves malicious content?
- Why are separate syntax-error and runtime-error judges proposed in the evaluation pipeline? Most syntax errors can be easily corrected by a coding agent with a linter as one of its tools.
- The “Problem Definition” section of the paper feels somewhat overly lengthy, since most of the formalisms and terminology introduced is not used elsewhere in the paper.

---

> ### Author Response · Authors · 2025-11-20
> **Author Reply (Part 1)**
>
> Thank you to the reviewer for their constructive comments and for acknowledging the strengths of our work, including the different workspace levels, the intuitive judge framework, and the interesting insights it enables.
>
>
> > “2 splits of the codebase is derived from existing work, and no analysis is provided for the quality of the CAJ-M split, which was created by this work. Therefore, the proposed benchmark largely builds on existing work. How does CAJ compare with other coding jailbreak benchmarks in terms of scale?”
>
> Thank you for this thoughtful comment. We agree that CAJ-0 reuses prompts from RMCBench, and this is **intentional**: it allows us to anchor our results to an existing, well-studied malicious code benchmark while shifting the setting from base LLMs to autonomous code agents. The main dataset novelty lies in (i) how we **restructure this material into agent-centric regimes** (empty vs single-file vs multi-file workspaces) and (ii) the **new CAJ-1 and CAJ-M splits**, where we construct single-file and multi-file malicious repositories specifically designed for end-to-end execution and evaluability within code agents.
> On the quality of CAJ-M, we agree that additional analysis strengthens the benchmark. Therefore, for this rebuttal, we have leveraged **VirusTotal’s Code Insight API and CodeQL analysis, along with manual inspection**, to verify that repositories implement the intended malicious behaviors.
> We want to emphasize that – CAJ is the first benchmark in jailbreaking code agents that explores multi-level workspaces, and mirrors real-life attackers’ capabilities. Our **CAJ-0 contains 182 jailbreaking prompts** consisting of direct and indirect approaches, **CAJ-1 has 100 samples** with partial single-file repository, and **CAJ-M consists of 180 repositories** having malicious programs in multiple files. Each sample corresponds to a full autonomous agent run (multiple tool calls, edits, and executions) evaluated through our four-stage pipeline (refusal, attack intent, syntax, and runtime). This yields a substantial number of trajectories and executed artifacts **across 11 attack categories, 3 regimes, and 7 models**, and we observe consistent trends across all of them. The key goal of this work is to **establish and validate the framework and show clear, reproducible escalations in risk as code context increases, rather than to maximize scale**. We will clarify this framing and also note that, upon release, our tooling and CAJ-Bench design are intended to support future work that scales to larger datasets and additional sources.
>
> > “No analysis is provided to show whether the refusal judge and attack evaluation judge results in human-aligned evaluation of ASRs.”
>
> Thank you for this constructive suggestion. In response, we have **incorporated a human evaluation** to assess how well the Refusal Judge and Attack Evaluation Judge align with human judgments of ASR.
> Concretely, we randomly sampled 100 agent-generated artifacts across settings and had **multiple authors manually annotate them** for (i) refusal vs. compliance and (ii) harmful vs. non-harmful intent/function. We then compared these labels with our LLM-based judges:
>
> * For CAJ-0, we observed only 2 cases where humans marked an artifact as unclear but the Attack Evaluation Judge labeled it harmful, and 1 case where the judge labeled it unclear but humans labeled it harmful.
> * For CAJ-1 and CAJ-M, human labels and judge verdicts matched on all sampled instances for both refusal and attack evaluation.
>
>
> We will add this human-alignment analysis to the revised manuscript to better substantiate that our ASR measurements are closely aligned with human judgments. Thanks again for this feedback.
>
> > “While this work focuses on executability of generated code, it does not consider how easily the generated code can be debugged via an agentic framework to be made executable, as long as it encodes the malicious intent. This somewhat limits the practicality of this benchmark, in my opinion.”
>
>
> Thank you for this insightful comment. We would like to clarify that, in our setup, the agent **already performs iterative debugging** as part of the generation process. OpenHands is allowed to run in a fully autonomous, **multi-step manner**: it can edit files, re-run code, inspect errors, and refine its implementation until it decides the task is complete. The executability metrics we report (syntax- and runtime-error-free rates) are therefore measured **after this agentic refinement loop, not after a single-shot code generation**.
> This is why, for stronger models such as GPT-4.1, GPT-o1, DeepSeek-R1, and Qwen3, we observe relatively high conversion from non-refusal to executable malicious code (see Figure 2). Please find some of the examples under Appendix A.2 (Figures 10-13), where we show the high-quality malicious code generated and debugged by these code agents in an iterative manner.

---

> > ### Author Response · Authors · 2025-11-20
> > **Author Reply (Part 2)**
> >
> > > “While the paper claims that LLMs are more vulnerable when wrapped in an agentic framework, the only framework used in the paper is the OpenHands framework. An agentic framework with explicit checks for malicious intent would be more resilient to attacks than the base LLM.”
> >
> > Yes, indeed. We agree with you, and this is **one of the main messages** of our work. The code agents we study, in their current state, do not incorporate strong safety guardrails, and **we show why such guardrails are necessary**. In particular, using OpenHands as a representative framework, we analyze the trajectories and identify how and when the agent’s behavior starts deviating from that of the base LLM (see Figure 8).
> > We believe this work helps establish a foundation for secure and safe programming in code-agent settings, and we fully agree that future agentic frameworks with explicit checks for malicious intent could be more resilient than the base LLM. Our hope is that our analysis will make enterprises and organizations more mindful of the guardrails they build into their code-agent frameworks.
> >
> > > “Can this benchmark be used to evaluate Claude Code/Codex-style agentic frameworks?”
> >
> > Yes, absolutely. Our choice of OpenHands in the initial submission was primarily **pragmatic**: it provides a mature, fully open-source code agent with strong tooling and easy integration into our multi-judge evaluation pipeline. That said, our methodology (CAJ-Bench + executability-aware judges) is designed to be **agent-agnostic**, and can be plugged into other scaffolds with minimal changes.
> > In response to your comment, we have already begun running experiments with **SWE-Agent** and **OpenAI Codex Agent** on CAJ-0. For GPT-4.1, preliminary results are very similar to those observed with OpenHands, suggesting that the core trends we report (e.g., high ASR and substantial rates of executable malicious code under simple attacks) are **not specific to a single agent framework**. We will incorporate these SWE-Agent and Codex Agent results into the revised version (space permitting) and explicitly discuss how our framework can be applied to additional agents going forward.
> >
> > | CAJ-0              | Evaluation Metric       | SWE-Agent | Codex (OpenAI) | OpenHands |
> > |--------------------|------------------------|-----------|----------------|-----------|
> > | Explicit Prompting | Compliance Rate        | 33.75%    | 22.50%         | 15.00%    |
> > |  | +Attack Success Rate   | 28.75%    | 16.25%         | 15.00%    |
> > | Implicit Prompting | Compliance Rate        | 44.12%    | 55.88%         | 50.98%    |
> > |  | +Attack Success Rate   | 33.33%    | 37.25%         | 49.02%    |
> > | All                | Compliance Rate        | 39.56%    | 41.21%         | 32.99%    |
> > |                | +Attack Success Rate   | 31.32%    | 28.02%         | 32.01%    |
> >
> >
> >
> > > “Does the executability judge approach introduced in this work generalize to other coding jailbreak benchmarks?”
> >
> > Yes, it does. Our executability-judge framework is **benchmark-agnostic**: it does not depend on which agent or dataset is used. The only inputs it needs are (i) the initial prompt and (ii) the final codebase/directory produced by the agent. Given these, the robustness and executability checks can be run on any coding jailbreak benchmark.
> > We have implemented this in a very modular way. In the supplementary material, we include two folders, ‘`robustness_judge`’ and ‘`exec_judge`’, containing the necessary Python and shell scripts. These are designed to be **plug-and-play** and can be **adapted directly** to other coding jailbreak benchmarks.

---

> > > ### Author Response · Authors · 2025-11-20
> > > **Author Reply (Part 3)**
> > >
> > > > “In CAJ-1/CAJ-M, are checks done to ensure that the part of the file that the agent is tasked with filling in also involves malicious content?”
> > >
> > > In CAJ-1 and CAJ-M, we start from codebases that are malicious as a whole, but we do not enforce that the \<FILL\_HERE\> placeholder always replaces the core malicious payload itself. The hollowed region can be malicious or non-malicious in isolation (e.g., helper logic, glue code, or part of the control flow), while the **overall repository remains malicious**.
> > > This is **intentional and tied to our threat model**: we want to test whether a code agent, when placed inside an already malicious project, will recognize the malicious context and choose to refuse, rather than blindly complete the repo. In other words, the benchmark is not designed so that the filled span alone must be malicious; instead, we evaluate the final completed codebase using our Attack Evaluation Judge (plus VirusTotal / CodeQL and human checks in our updated analysis) to **determine whether the resulting artifact implements malicious behavior**.
> > >
> > > > “Why are separate syntax-error and runtime-error judges proposed in the evaluation pipeline? Most syntax errors can be easily corrected by a coding agent with a linter as one of its tools.”
> > >
> > > Great question; thanks for raising it. We want to gently remind that our syntax-error and runtime-error judges are not simple string checks, but **agentic components**, as described in subsection 4.2. The syntax-error judge runs a small toolchain (e.g., tree-sitter, py_compile, etc.) over the final codebase, and the runtime-error judge uses a non-mutating micro-agent to plan and execute the code in a sandboxed environment.
> > > **The reason we keep syntax and runtime as separate stages is to distinguish between different notions of “harmful” code**:
> > >
> > > * The Attack Evaluation Judge says: this code is malicious in intent/function.
> > > * The Syntax-Error Judge checks: does this malicious code even parse/compile?
> > > * The Runtime-Error Judge then checks: does it actually run end-to-end?
> > >
> > >
> > > Prior work would typically stop at the first step, and label all such cases as “harmful”, even if they never execute successfully. We show that a non-trivial fraction of artifacts **fall into this gap: seemingly harmful, but undeployable due to syntax or runtime issues** (such examples in Figures 12 and 21). Our separation of syntax and runtime makes this gap explicit and yields a more nuanced understanding of deployable harm.
> > >
> > > > “The “Problem Definition” section of the paper feels somewhat overly lengthy, since most of the formalisms and terminology introduced is not used elsewhere in the paper.”
> > >
> > > Thanks for this constructive feedback. We tried to keep the Problem Definition section concise and clearly convey our **threat model and the attacker’s capabilities**, and show how we keep the attacks realistic and intuitive. That said, we agree that some of the formalisms are not strictly necessary for following the rest of the paper.
> > > Based on your comment, we are revising this section to make it more concise, keeping only the notation and definitions that are actually used later in the benchmark and analysis. Thanks again for raising this.

---

> > > > ### Author Response · Authors · 2025-11-21
> > > > **Paper Revision + Author Reply**
> > > >
> > > > Dear Reviewer,
> > > >
> > > > We are glad to let you know that we have carefully incorporated your suggestions into the revised version of the paper. In the uploaded revised PDF (edits marked in blue), the main changes addressing your comments are:
> > > > * **Human alignment of robustness judges:** We performed human verification of the Refusal and Attack Evaluation Judges and added the results as a dedicated paragraph in Section 4.1.
> > > > * **Agent-agnostic ablation:** We added an ablation study extending CAJ-Bench to other code agents (SWE-Agent and a Codex-style agent from OpenAI) on CAJ-0, and report the results in Section 7 under “Extending to Different Code Agents.”, showing that our benchmark is agent-agnostic.
> > > > * **Judge pipeline clarification:** We introduced a new subsection, Section 4.3 (“Judge Framework”), to clearly describe how the four-stage judge pipeline (refusal → attack evaluation → syntax → runtime) is composed, emphasizing that the syntax and runtime judges are only applied to artifacts already labeled harmful. We also explain how the `robustness_judge` and `exec_judge` components can be used in a plug-and-play fashion with other benchmarks and agents.
> > > > * **CAJ-M quality validation:** We added validation of CAJ-M using VirusTotal Code Insight and CodeQL, together with manual inspection, in the last paragraph of Section 3 (CodeAgentJail Benchmark).
> > > > * **More concise Problem Definition:** We have shortened and streamlined the “Problem Definition” section to retain only the notation and assumptions.
> > > >
> > > > We hope these revisions address your concerns. Thank you again for your detailed and constructive feedback, which we believe has significantly improved the paper. Please let us know if you have any further questions or concerns.

---

> > > > > ### Comment · Reviewer_z1ru · 2025-11-25
> > > > >
> > > > > I appreciate the authors’ thorough responses in the rebuttal and have raised my score accordingly. The main reason I am not able to give the paper an 8 is that, although the executability metric forming the core of this work is relevant and well-motivated, I believe it still does not fully capture the attack surface. In practice, frameworks like OpenHands often produce code that becomes executable with minimal manual debugging, which is the pattern that closely reflects real-world use of these coding agents. For instance, when the syntax-error judge flags issues for some executed code, these can often be resolved simply by feeding the error traces back into the agentic coding model. The resulting code will still encode some malicious intent. As a result, I am still uncertain about how much additional value this work provides beyond existing benchmarks that just assess maliciousness or refusal rates.

---

### Official Review · Reviewer_rYaq · 2025-11-01

**Soundness:** 3
**Presentation:** 3
**Contribution:** 2
**Rating:** 6
**Confidence:** 4

**Summary:**

This paper introduces CODEAGENTJAIL, a benchmark and evaluation framework for assessing the security vulnerabilities of code-capable LLM agents under systematic jailbreak attacks. Unlike prior jailbreak studies focused on textual refusal, this work measures whether code agents actually generate and execute malicious code within realistic workspaces. The authors define three escalating attack regimes: CAJ-0 (empty workspace, prompt-only), CAJ-1 (single-file partial code), and CAJ-M (multi-file repository), and design a four-phase judge pipeline that evaluates compliance, harmfulness, syntactic correctness, and runtime executability. Experiments on seven major LLM families within the OpenHands agent framework show that (1) code agents are substantially more vulnerable than base LLMs (≈1.6× ASR increase), (2) compliance approaches 100% in seeded workspaces, and (3) up to 32% of multi-file attacks produce deployable malicious programs. The study highlights critical weaknesses in current safety mechanisms and calls for execution-aware and refusal-persistent defenses.

**Strengths:**

The benchmark is methodologically innovative, bridging safety evaluation and systems engineering. Its multi-stage structure cleanly separates intent from operational harm, offering a more realistic safety metric. The experiments are large-scale, multi-model, and reproducible under open-source infrastructure. The ablation contrasting base LLMs and agentic wrappers compellingly isolates the cause of vulnerability escalation. The paper also contributes thoughtful discussions and future research directions on refusal persistence and execution-aware safety.

**Weaknesses:**

While the paper is technically rigorous and methodologically sophisticated, several limitations constrain its novelty claims and generalizability.

1. Overstated novelty of the “code-format vulnerability” finding.
The paper highlights that malicious requests written in code or tool-like formats are more likely to be executed than refused. However, this phenomenon has already been systematically demonstrated in prior literature, most notably in Drop the Guardrails: Tool-Primed Prompt Pairing and Refusal Behavior in GPT-OSS (Kevin Power, 2025; Kaggle Write-up, OpenAI Red-Teaming Hackathon Winner). That study quantified 15–42 percentage-point reductions in refusal rates when harmful requests were framed as code or tool calls, providing direct empirical support for the same effect discussed here.

2. Limited language and environment diversity.
All evaluations are conducted in Python-based Dockerized workspaces using OpenHands agents. While this ensures reproducibility, it limits ecological validity since real-world code agents frequently operate in polyglot settings (e.g., JavaScript, SQL, shell scripting).

3. Interpretability of “execution success.”
Treating runtime executability as the primary indicator of harm can overstate actual exploitability. Many successful runs may simply produce syntactically valid but non-dangerous programs. Incorporating static vulnerability scanning (e.g., CodeQL or Semgrep) or privilege-aware evaluation would provide a more nuanced view of operational risk.

**Questions:**

1. How exactly does this work extend Power (2025)’s “tool/code” findings? Is the main novelty the runtime/executable evaluation or something else?
2. Can you show an example trace where an initial refusal is overturned during the agent’s planning/tool steps?
3. How are Docker sandboxes configured to prevent real harm (network egress, privileges)?
4. Did you try any simple defenses (e.g., persist refusal state across steps, pre-execution safety gate)?
5. How does “runtime-error-free” map to real-world risk? Do you plan to add static checks (CodeQL/Semgrep) or human verification to filter false positives?

---

> ### Author Response · Authors · 2025-11-20
> **Author Reply (Part 1)**
>
> Thank you for your constructive feedback and for recognizing the strengths of our benchmark and framework, including their methodological design and the way they bridge safety evaluation with executable, systems-level behavior.
>
>
> > “Overstated novelty of the “code-format vulnerability” finding. The paper highlights that malicious requests written in code or tool-like formats are more likely to be executed than refused. However, this phenomenon has already been systematically demonstrated in prior literature, most notably in Drop the Guardrails: Tool-Primed Prompt Pairing and Refusal Behavior in GPT-OSS (Kevin Power, 2025; Kaggle Write-up, OpenAI Red-Teaming Hackathon Winner). That study quantified 15–42 percentage-point reductions in refusal rates when harmful requests were framed as code or tool calls, providing direct empirical support for the same effect discussed here.”
>
> Thanks for bringing this work to our attention. We want to politely remind the reviewer that, according to the ICLR guidelines (https://iclr.cc/Conferences/2026/ReviewerGuide), any paper that was published after July 24, 2025 is considered *contemporaneous*, and as authors we are not **required to compare** against such work in the submission.
>
> That said, we want to point out that, in our CAJ-Benchmark, we already perform a **rigorous and fine-grained analysis** of the “malicious requests written in code” phenomenon by explicitly designing different levels of workspaces (empty, single-file, multi-file) and studying how code-format and code context affect refusal, attack success, and executability.
>
> In our revised version, we will refer to *'Drop the Guardrails'* as concurrent work and clearly state how our results are consistent with, and further support, their findings.
>
> > “Limited language and environment diversity. All evaluations are conducted in Python-based Dockerized workspaces using OpenHands agents. While this ensures reproducibility, it limits ecological validity since real-world code agents frequently operate in polyglot settings (e.g., JavaScript, SQL, shell scripting).”
>
> Thanks for bringing this and giving us the opportunity to clarify. While the majority of the code generated by the Code Agents is indeed Python, we want to confirm that our **benchmark contains other languages**. For example, the PID 61 and 62 in our CAJ-0 explicitly prompt to generate malicious code in JavaScript (we have added the dataset in the supplementary material). Moreover, in terms of language choice, we **give the agent full freedom** since some of the attacks might require a specific language implementation. For example, for PID 31 in CAJ-0 (“generate a simple process hollowing example”), the jailbroken agent generates a code in C language and adds that as a payload.
>
> So, yes, you are right, and we agree with you that in real-world code agents operate in polyglot settings. We will explicitly mention this in our revised manuscript for better understanding. Thanks again for pointing this out.

---

> ### Author Response · Authors · 2025-11-20
> **Author Reply (Part 2)**
>
> > “Interpretability of “execution success.” Treating runtime executability as the primary indicator of harm can overstate actual exploitability. Many successful runs may simply produce syntactically valid but non-dangerous programs. Incorporating static vulnerability scanning (e.g., CodeQL or Semgrep) or privilege-aware evaluation would provide a more nuanced view of operational risk.”
>
> > “How does “runtime-error-free” map to real-world risk? Do you plan to add static checks (CodeQL/Semgrep) or human verification to filter false positives?”
>
> Thanks for raising this point. We fully agree that “runtime-error-free” alone does not guarantee real-world exploitability, and this was exactly the **motivation for designing our multi-stage judge framework rather than treating executability as a standalone signal**.
> As a reminder, our pipeline is:
>
> 1. Refusal Judge
> 2. Attack Evaluation Judge (malicious vs. non-malicious intent/function)
> 3. Syntax-Error Judge
> 4. Runtime-Error Judge
>
>
> Crucially, the Syntax-Error and Runtime-Error judges are **only invoked** if the Attack Evaluation Judge **first labels the artifacts as harmful**. In other words, we do not evaluate syntax/runtime correctness on programs that are judged non-dangerous. The reported **“+ Runtime-Error-Free”** numbers therefore map to: “prompts for which the agent complied and produced artifacts judged as harmful and those harmful artifacts parse and run successfully in our sandbox.”
>
> In response to your suggestion, we have also begun incorporating static analysis to refine our notion of operational risk. Specifically, for this rebuttal, we have leveraged **VirusTotal’s Code Insight API** and **CodeQL analysis**, along with manual inspection, to verify that repositories implement the intended malicious behaviors.
>
> Additionally, we have also **incorporated a human evaluation** to assess how well the Refusal Judge and Attack Evaluation Judge align with human judgments of ASR. Concretely, we randomly sampled 100 agent-generated artifacts across settings and had **multiple authors manually annotate** them for (i) refusal vs. compliance and (ii) harmful vs. non-harmful intent/function. We then compared these labels with our LLM-based judges:
>
> * For CAJ-0, we observed only 2 cases where humans marked an artifact as unclear but the Attack Evaluation Judge labeled it harmful, and 1 case where the judge labeled it unclear but humans labeled it harmful.
> * For CAJ-1 and CAJ-M, human labels and judge verdicts matched on all sampled instances for both refusal and attack evaluation.
>
>
> We believe these additions – static CodeQL and VirusTotal Code Insight checks plus human validation – should give the community more confidence in the reliability of our benchmark. Thanks for this feedback.
>
> > “How exactly does this work extend Power (2025)’s “tool/code” findings? Is the main novelty the runtime/executable evaluation or something else?”
>
> As we mentioned earlier, we would request the reviewer to consider this work as *‘contemporaneous’* according to the reviewer guidelines. However, for clarity, these are the ways our work differs from the Power (2025) work:
>
> 1. **Attack Surface:** Powel (2025) focuses on textual prompts paired with a fake tool manifest. Each harmful task is asked in plain‑text and with structured API‑like syntax to see how tool‑priming shifts refusal. In contrast, CAJ-Bench emphasises executable code context: three workspace regimes (empty, single‑file, multi‑file) that reflect naive, capable and expert attackers. The agent must complete or generate code, with multi‑step planning and real tool calls.
> 2. **Tool Handling:** While Powel (2025) injects a notional tool manifest into the system prompt, in our work, the attacker cannot modify the tool list; instead, the attack proceeds purely by exploiting how the agent behaves when placed in different code contexts (empty vs. partial vs. multi-file malicious repos).
> 3. **Evaluation Pipeline:** Powel (2025) measures only refusal vs. compliance. On the other hand, we design a sophisticated evaluation pipeline. We design a hierarchical, four‑phase judge that measures compliance, attack success (harmfulness), syntax correctness, and runtime executability. This lets us quantify not just whether “tool/code-like” formatting reduces refusals, but how often it leads all the way to syntactically valid and runtime-successful malicious programs in realistic agent workspaces.
>
> So, the novelty is not only the runtime/executable evaluation, but also the multi-level, workspace-based attack model and the fine-grained analysis of added code context in autonomous code agents, which we see as a natural extension of the phenomenon observed in Power (2025) into a more realistic agentic and code-centric setting.

---

> > ### Author Response · Authors · 2025-11-20
> > **Author Reply (Part 3)**
> >
> > > “Can you show an example trace where an initial refusal is overturned during the agent’s planning/tool steps?”
> >
> > Yes. Upon our manual inspection of the agent trajectories and log files, we do find a recurring pattern where the agent’s behavior changes during a multi-turn interaction. An example is already provided in **Appendix A.1 (Figure 8)**: the attacker prompts for a DDoS attack, and the LLM (GPT-4.1) initially refuses to comply. However, after the pre-defined “helping” prompt from the agent, the LLM invokes its thinking tool and then decides to provide code. What starts as an “educational demo” gradually turns into genuinely malicious code over subsequent steps. We describe this pattern in more detail in Section 7 (Ablation Study).
> >
> >
> >
> >
> > > “How are Docker sandboxes configured to prevent real harm (network egress, privileges)?”
> >
> > Thanks for raising this concern. All code is executed inside isolated Docker containers with (i) no outbound network access and (ii) non-privileged, non-root users and minimal filesystem access, so the code cannot reach external services or escalate privileges on the host. In addition, our runtime executability judge is configured as a non-mutating micro-agent based on the OpenHands (June) framework: unlike the main code agent, it is not given any file-editing or write access – only read/inspection and run capabilities inside the container. Combined, this ensures that any malicious behavior remains contained in the sandbox while still allowing us to faithfully assess executability.
> >
> >
> > > “Did you try any simple defenses (e.g., persist refusal state across steps, pre-execution safety gate)?”
> >
> > Thanks for this question. The main goal of this work was to show the existing and realistic scope of exploitation in code agents **in their current state**, and hence, we kept our focus on the attack side rather than on adding new defenses. As we mention in our “Future Work” section, we view our multi-judge framework as a natural backbone for such defenses (e.g., using the attack/exec judges as a pre-execution gate or enforcing persistent refusal across planning steps), and we believe our results will encourage the community and developers to incorporate these stronger safety guardrails into future code-agent designs.

---

> > > ### Author Response · Authors · 2025-11-21
> > > **Paper Revision + Author Reply**
> > >
> > > Dear Reviewer,
> > >
> > > We are glad to let you know that we have carefully incorporated your suggestions into the revised version of the paper. In the uploaded revised PDF (edits marked in blue), the main changes addressing your comments are:
> > > * **Connection to “Drop the Guardrails”:** We now explicitly state that our finding that added code context helps jailbreaking is consistent with Power (2025), and we position that work as concurrent to ours in subsection 6.3.
> > > * **Language diversity:** We clarify that CAJ-Bench contains samples in languages beyond Python, and that agents are free to choose any language when completing tasks. These details are discussed in Appendix A.3 (CodeAgentJail-Bench Details).
> > > * **Judge pipeline clarification:** We introduced a new subsection, Section 4.3 (“Judge Framework”), to clearly describe how the four-stage judge pipeline (refusal → attack evaluation → syntax → runtime) is composed and to emphasize that syntax and runtime judges are only applied to artifacts already labeled harmful.
> > > * **CAJ-M quality validation:** We added validation of CAJ-M using VirusTotal Code Insight and CodeQL, together with manual inspection, in the last paragraph of Section 3 (CodeAgentJail Benchmark), to further substantiate that CAJ-M repositories implement their intended malicious behaviors.
> > > * **Human alignment of robustness judges:** We performed human verification of the Refusal and Attack Evaluation Judges and added the results as a dedicated paragraph in Section 4.1.
> > > * **Docker sandbox details:** We clarified how the Docker sandbox and executability judge are configured in Section 4.2, highlighting that the executability judge is a non-mutating micro-agent operating on an isolated workspace inside Docker.
> > >
> > > In addition, we made the following improvements:
> > > * **Agent-agnostic ablation:** We added an ablation study extending CAJ-Bench to other code agents (SWE-Agent and a Codex-style agent from OpenAI) on CAJ-0, and report the results in Section 7 under “Extending to Different Code Agents.”,  showing that our benchmark is agent-agnostic.
> > > * **Extensions to attacks and defenses:** In the Conclusion, we now explicitly state that CAJ-Bench and the Judge Framework can be extended to more sophisticated attacks and execution-aware defenses.
> > >
> > > We hope these revisions address your concerns. Thank you again for your detailed and constructive feedback, which we believe has significantly improved the paper. Please let us know if you have any further questions or concerns.

---

### Official Review · Reviewer_UA7N · 2025-11-03

**Soundness:** 2
**Presentation:** 2
**Contribution:** 1
**Rating:** 4
**Confidence:** 3

**Summary:**

This paper presents CODEAGENT JAIL-Bench, a benchmark designed to assess the security of code-capable LLM agents against jailbreaking attacks across three increasingly realistic workspace regimes: empty (CAJ-0), single-file (CAJ-1), and multi-file (CAJ-M). The authors introduce a hierarchical, four-phase judge framework that evaluates not only policy compliance and attack success but also the syntactic correctness and runtime executability of the generated code, thereby measuring deployable harm rather than just textual refusal. Evaluations of seven LLMs reveal that wrapping models in an agentic framework increases attack success rates by an average of 1.6x, as multi-step planning often overturns initial refusals. The study finds that while single-file contexts drive compliance to nearly 100% for capable models, the multi-file regime is the most dangerous, yielding 32% instantly deployable attack code.

**Strengths:**

- The benchmark introduces three escalating regimes (CAJ-0, CAJ-1, CAJ-M) that mirror real-world attacker capabilities, ranging from naive prompt-only attacks to single-file to multi-file repository manipulation.
- This framework includes LLM judges to test if the malicious code actually parses, builds, and runs to completion, providing a strong measure of executable harm.
- The paper provides empirical evidence that code agents are more vulnerable than their base LLMs as iterative reasoning and tool feedback loops can erode initial safety decisions.

**Weaknesses:**

- Technical Novelty in Jailbreaking: Despite using "jailbreak" in the title, the paper does not any automated jailbreaking algorithms such as gradient-based or genetic search methods. It relies instead on direct prompting strategies and curated malicious codebases.
- Security Scope: The evaluation primarily focuses on malware generation across categories like spyware and ransomware, without considering other critical security issues such as Common Weakness Enumeration (CWE) vulnerabilities.
-  Dataset Contribution & Size: The CAJ-0 dataset is derived from existing benchmarks (RMCBench), limiting its novelty. The datasets are also relatively small: CAJ-0 contains only 182 prompts , CAJ-1 has 100 samples , and CAJ-M consists of 180 repositories
- Evaluation:
  - Deployable Harm: While the "Runtime-Error Judge" checks if code executes, it is unclear if the code is genuinely harmful or capable of causing real-world attacks.
  - Reliance on LLM Judges: Attack success is determined by an LLM judge (specifically Claude-3.7-Sonnet ), which may suffer from inaccuracies compared to ground-truth security assessments. The authors could consider utilizing external/commercial validation tools like VirusTotal to confirm the detectability or functional maliciousness of generated code.
  - Compared to Prior Work: The conclusion that agents are more harmful than base LLMs has been previously explored in works like RedCode (Guo et al., 2024a), raising questions about the extent of new insights provided by this specific experimental setup.

- Agent Framework and Model Diversity:
  - The evaluations only use the OpenHands agent framework, overlooking other relevant scaffolds like SWE-agent.
  - High syntax and runtime error rates (e.g., only 4.33% average runtime success in CAJ-1 ) might stem from the base models' limited coding capabilities rather than effective safety guardrails. The evaluated models appear somewhat weak, and the inclusion of more state-of-the-art models (e.g., closed-source models) could be considered.

**Questions:**

See weaknesses part.

---

> ### Author Response · Authors · 2025-11-20
> **Author Reply (Part 1)**
>
> Thanks to the reviewer for their constructive comments and for acknowledging the strengths of our work, such as mirroring real-world attacker capabilities and providing a stronger measure of executable harm.
>
>
> > “Technical Novelty in Jailbreaking: Despite using "jailbreak" in the title, the paper does not any automated jailbreaking algorithms such as gradient-based or genetic search methods. It relies instead on direct prompting strategies and curated malicious codebases”
>
>
> Thank you for this comment – we agree that there is an important line of work on automated jailbreak generation (e.g., gradient-based or evolutionary search), and our work is **complementary to that direction** rather than competing with it.
> While such algorithms have shown a higher attack success rate, the goal of our work is not to optimize the jailbreak attack, but rather to **show that existing code agents are vulnerable under simpler attacks**. We therefore deliberately focus on direct/indirect prompting plus realistically structured malicious codebases, which correspond to what **a “naive” or “capable” attacker can do without access to optimization infrastructure**.
> Even under this conservative threat model, we already observe:
>
> * In CAJ-0, direct prompting alone yields an average **ASR of ~58%, which is high given there is no search or optimization**.
> * Our indirect prompting further amplifies **ASR by up to 3.45×** over direct prompting for some models, and many of these attacks produce fully executable malicious code.
>
> These results suggest that **current code agents are vulnerable even before considering sophisticated automated jailbreaks**. We see CAJ-Bench and our executability-aware evaluation pipeline as **a foundation on which future work can plug in gradient-based/genetic search methods** to explore stronger attacks and more advanced defenses. We will clarify this positioning and the rationale for our threat model in the revised version.
>
> > “Security Scope: The evaluation primarily focuses on malware generation across categories like spyware and ransomware, without considering other critical security issues such as Common Weakness Enumeration (CWE) vulnerabilities.”
>
> Thank you for highlighting this scope question. The primary goal of this work is to show how an attacker can exploit code agents to generate and deploy malicious code. While CWE vulnerabilities are another scope of security, they are not **deliberate exploitation like a jailbreak**, but rather subtle insecure patterns in code. In this paper, we keep our focus narrow to ‘jailbreaking code agent to generate malicious programs’ and deep dive by exploring different levels of workspace complexity in this scope itself.
>
> > “Dataset Contribution & Size: The CAJ-0 dataset is derived from existing benchmarks (RMCBench), limiting its novelty. The datasets are also relatively small: CAJ-0 contains only 182 prompts , CAJ-1 has 100 samples , and CAJ-M consists of 180 repositories.”
>
> Thank you for this thoughtful comment. We agree that CAJ-0 reuses prompts from RMCBench, and this is **intentional**: it allows us to anchor our results to an existing, well-studied malicious code benchmark while shifting the setting from base LLMs to autonomous code agents. The main dataset novelty lies in (i) how we **restructure this material into agent-centric regimes** (empty vs single-file vs multi-file workspaces) and (ii) the **new CAJ-1 and CAJ-M splits**, where we construct single-file and multi-file malicious repositories specifically designed for end-to-end execution and evaluability within code agents.
> Regarding size, while 182/100/180 may seem modest in absolute terms, each sample corresponds to **a full autonomous agent run (multiple tool calls, edits, and executions) evaluated through our four-stage pipeline (refusal, attack intent, syntax, and runtime)**. This yields a substantial number of trajectories and executed artifacts **across 11 attack categories, 3 regimes, and 7 models**, and we observe consistent trends across all of them. We want to emphasize again that – the key goal of this work is to **establish and validate the framework and show clear, reproducible escalations in risk as code context increases**, rather than to maximize scale. We will clarify this framing and also note that, upon release, our tooling and CAJ-Bench design are intended to support future work that scales to larger datasets and additional sources.

---

> > ### Author Response · Authors · 2025-11-20
> > **Author Reply (Part 2)**
> >
> > > “Deployable Harm: While the "Runtime-Error Judge" checks if code executes, it is unclear if the code is genuinely harmful or capable of causing real-world attacks.”
> >
> > Thank you for raising this important point. In our framework, the Runtime-Error Judge is **only applied after** the Attack Evaluation Judge has already labeled the produced artifacts as malicious in intent and functionality. In other words, the reported **“+ Runtime-Error-Free Rate”** corresponds to the fraction of prompts for which (i) the agent complied, (ii) the generated artifacts are judged as harmful, and (iii) those harmful artifacts parse and execute without errors in our sandbox.
> > We fully agree that actually deploying malware “in the wild” would be the gold standard for measuring real-world harm, but this is neither practical nor ethical. Our goal is therefore to **approximate deployable harm** as: code that implements malicious logic and is directly runnable end-to-end in a realistic, but controlled, environment. We will clarify this definition in the paper and emphasize that our multi-judge pipeline is designed precisely to **go beyond “it runs” to “it runs and has been independently judged as malicious”**, while still operating within safe experimental constraints.
> >
> > > “Reliance on LLM Judges: Attack success is determined by an LLM judge (specifically Claude-3.7-Sonnet ), which may suffer from inaccuracies compared to ground-truth security assessments. The authors could consider utilizing external/commercial validation tools like VirusTotal to confirm the detectability or functional maliciousness of generated code.”
> >
> > Thanks for this suggestion. While we have already leveraged human verification on the Judge verdicts, we agree that accompanying a commercial validation tool can strengthen our claim. Hence, as per your suggestions, we have integrated the **VirusTotal’s Code Insights API** and **CodeQL analysis**, and are running the experiments. We will add these results to our main paper (space permitting) and report the results to you before the rebuttal period ends. Thanks again for this feedback.
> >
> > > “Compared to Prior Work: The conclusion that agents are more harmful than base LLMs has been previously explored in works like RedCode (Guo et al., 2024a), raising questions about the extent of new insights provided by this specific experimental setup.”
> >
> > Thank you for pointing out the connection to RedCode (Guo et al., 2024a). We agree that the high-level observation that agents can be more harmful than base LLMs has been reported before, and we do not intend to present this statement itself as a novel contribution. In our work, this comparison primarily serves as a **sanity check that our findings align with prior literature**.
> > Besides this, we have provided other significant insights as well, such as, **subsection 6.1:** model’s tendency to generate harmful code once they comply, implicit/indirect prompting achieves higher ASR; **subsection 6.2:** single-file workspace elevates ASR for strong/robust models; **subsection 6.3:** highest executable malicious code generation in multi-file workspace, adding code-context helps in jailbreaking; **section 7:** varying jailbreak success in different malicious categories, etc. Moreover, we want to emphasize that we did **trajectory-level analysis** to find the underlying reasoning for agents' higher vulnerability than base LLMs, and **identify the turning points** where the agents start differing from the base LLMs (Figure 8).
> > We want to tone down the finding “agents being more harmful than base LLMs” and reframe it as an alignment with the work RedCode (Guo et. al. 2024) in our revised manuscript.

---

> > > ### Author Response · Authors · 2025-11-20
> > > **Author Reply (Part 3)**
> > >
> > > > “The evaluations only use the OpenHands agent framework, overlooking other relevant scaffolds like SWE-agent.”
> > >
> > > Thanks for the valuable feedback. Our choice of OpenHands in the initial submission was **primarily pragmatic**: it provides a mature, fully open-source code agent with strong tooling and easy integration into our multi-judge evaluation pipeline. That said, our methodology (CAJ-Bench + executability-aware judges) is designed to be **agent-agnostic**, and can be plugged into other scaffolds with minimal changes.
> > > In response to your comment, we have begun running experiments with **SWE-Agent** and **OpenAI Codex Agent** on CAJ-0. For GPT-4.1, preliminary results are very similar to those observed with OpenHands, suggesting that the core trends we report (e.g., high ASR and substantial rates of executable malicious code under simple attacks) are **not specific to a single agent framework**. We will incorporate these SWE-Agent and Codex Agent results into the revised version (space permitting) and explicitly discuss how our framework can be applied to additional agents going forward.
> > >
> > > | CAJ-0              | Evaluation Metric       | SWE-Agent | Codex (OpenAI) | OpenHands |
> > > |--------------------|------------------------|-----------|----------------|-----------|
> > > | Explicit Prompting | Compliance Rate        | 33.75%    | 22.50%         | 15.00%    |
> > > |  | +Attack Success Rate   | 28.75%    | 16.25%         | 15.00%    |
> > > | Implicit Prompting | Compliance Rate        | 44.12%    | 55.88%         | 50.98%    |
> > > |  | +Attack Success Rate   | 33.33%    | 37.25%         | 49.02%    |
> > > | All                | Compliance Rate        | 39.56%    | 41.21%         | 32.99%    |
> > > |                | +Attack Success Rate   | 31.32%    | 28.02%         | 32.01%    |
> > >
> > >
> > >
> > >
> > > > “High syntax and runtime error rates (e.g., only 4.33% average runtime success in CAJ-1 ) might stem from the base models' limited coding capabilities rather than effective safety guardrails. The evaluated models appear somewhat weak, and the inclusion of more state-of-the-art models (e.g., closed-source models) could be considered.”
> > >
> > > Thank you for this insightful comment. We agree that high syntax and runtime error rates, especially in CAJ-1, can reflect limitations in base model coding capability and agent implementation, not just safety guardrails. In fact, we **explicitly discuss this in subsections 6.2 and 6.3**, and are careful not to interpret failures as evidence of effective safety.
> > > At the same time, we would respectfully push back on the characterization that the evaluated models are “weak.” Several of the models we use – GPT-4.1, GPT-o1, DeepSeek-R1, Qwen3-235B – are **among the top performers on standard coding benchmarks** such as BigCodeBench [1], MBPP, and HumanEval [2]. The very low runtime success rate in CAJ-1 therefore reflects a deliberately challenging single-file setting (tight constraints, missing imports/entrypoints, no explicit run harness), rather than simply poor coding ability. This is also **supported by our CAJ-M results**, where the same models achieve much higher syntax- and runtime-success once given a richer multi-file scaffold.
> > >
> > > [1] https://bigcode-bench.github.io/
> > >
> > > [2] https://evalplus.github.io/leaderboard.html

---

> > > > ### Author Response · Authors · 2025-11-21
> > > > **Paper Revision + Author Reply**
> > > >
> > > > Dear Reviewer,
> > > >
> > > > We are glad to let you know that we have carefully incorporated your suggestions into the revised version of the paper. In the uploaded revised PDF (edits marked in blue), the main changes addressing your comments are:
> > > >
> > > > * **Agent-agnostic ablation:** We added an ablation study extending CAJ-Bench to other code agents (SWE-Agent and a Codex-style agent from OpenAI) on CAJ-0, and report the results in Section 7 under “Extending to Different Code Agents.”,  showing that our benchmark is agent-agnostic.
> > > >
> > > >
> > > > * **CAJ-M quality validation:** We added validation of CAJ-M using VirusTotal Code Insight and CodeQL, together with manual inspection, in the last paragraph of Section 3 (CodeAgentJail Benchmark).
> > > >
> > > > * **Judge pipeline clarification:** We introduced a new subsection, Section 4.3 (“Judge Framework”), to clearly describe how the four-stage judge pipeline (refusal → attack evaluation → syntax → runtime) is composed and how syntax/runtime judges are only applied to artifacts already labeled harmful.
> > > >
> > > > * **Positioning vs. RedCode:** We toned down the claim that “agents are more harmful than base LLMs” and now explicitly frame this finding as consistent with RedCode (Guo et al., 2024) in the Introduction, Related Work, and Section 7.
> > > >
> > > > * **Extensions to attacks and defenses:** In the Conclusion, we now explicitly discuss how CAJ-Bench and the Judge Framework can be extended to more sophisticated attacks and execution-aware defenses.
> > > >
> > > > In addition, we have made the following improvements:
> > > >
> > > > * **Human alignment of robustness judges:** We performed human verification of the Refusal and Attack Evaluation Judges and added the results as a dedicated paragraph in Section 4.1.
> > > >
> > > > * **Docker sandbox details:** We clarified how the Docker sandbox and executability judge are configured (non-mutating micro-agent, isolated workspace) in Section 4.2.
> > > >
> > > > We hope these revisions address your concerns. Thank you again for your detailed and constructive feedback, which we believe has significantly improved the paper. Please let us know if you have any further questions or concerns.

---

### Author Response · Authors · 2025-12-03
**Author Final Remarks (1/3)**

## I. Acknowledgements
---
We would like to express our sincere gratitude to all reviewers for their thoughtful and constructive feedback, which significantly improved this work. We also thank the reviewers who increased their scores during the rebuttal period. Finally, we thank the Area Chair and the ICLR organizing committee for their effort in coordinating the review process and for their timely decision.

## II. Key Strengths
---
Reviewers highlighted strengths across four dimensions:

1. **Benchmark design & realism**
    * Reviewers emphasized that CAJ-Bench’s **three escalating workspace regimes (CAJ-0/CAJ-1/CAJ-M)** mirror real-world attacker capabilities, from prompt-only attacks to single-file and multi-file repository settings (UA7N, rYaq, z1ru, XHKK).
    * In particular, the **multi-file regime** was noted as both interesting and realistic (XHKK, UA7N).


2. **Executable-aware, multi-stage evaluation**
    * Reviewers noted that our **multi-stage judge framework** cleanly separates intent from operational harm and provides a more realistic safety metric than refusal-only evaluation (rYaq, z1ru, UA7N).
    * The inclusion of syntax and runtime checks to assess whether malicious code **parses/builds/runs end-to-end** was recognized as a strong measure of executable harm (UA7N, rYaq).


3. **Technical soundness, clarity, and reproducibility**
    * Reviewers found the paper **clearly written and easy to follow**, with an intuitive evaluation pipeline (XHKK, z1ru).
    * The experiments were described as **large-scale, multi-model, and reproducible** under open-source infrastructure (rYaq).


4. **Empirical insights on agentic vulnerability**
    * Reviewers highlighted that the paper provides empirical evidence and analysis showing **why agentic wrappers can amplify vulnerability**, including how iterative planning/tool feedback can overturn initial refusals (UA7N, rYaq, z1ru).
    * Reviewers also appreciated the **detailed analysis across malicious categories**, identifying which classes are most vulnerable in contemporary coding agents (z1ru).

---

> ### Author Response · Authors · 2025-12-03
> **Author Final Remarks (2/3)**
>
> ## III. Key Concerns and Our Responses
> ---
> Below, we summarize the major concerns raised by reviewers and how we addressed them in the rebuttal and revision. We were able to address all concerns raised by the reviewers through clarifications, additional analyses, and revisions. Consistent with our rebuttal commitments, all corresponding updates have been incorporated into the revised manuscript and are highlighted in blue in the uploaded PDF.
>
>
> | Major Concerns | Reviewers | Our Responses and Revision |
> |---|---|---|
> |**Generality beyond a single agent scaffold (OpenHands)** | UA7N, z1ru, XHKK | Added an **agent-agnostic ablation** by running CAJ-0 (GPT-4.1 backend) on **SWE-Agent** and **OpenAI Codex agent**, in addition to OpenHands, and reported results in **Section 7** (“Extending to Different Code Agents”), showing consistent trends across frameworks. |
> | **CAJ-M quality / malicious alignment / practicality** | UA7N, rYaq, z1ru, XHKK | Added **CAJ-M validation** using **VirusTotal Code Insight and CodeQL analysis**, together with **manual inspection**, and described this in the last paragraph of **Section 3** (CodeAgentJail Benchmark) to substantiate that CAJ-M repositories implement intended malicious behaviors. |
> | **Reliability of LLM judges (human alignment)** | rYaq, z1ru, XHKK | Added **human verification** of the **Refusal and Attack Evaluation** judges and reported the results as a dedicated paragraph in **Section 4.1**, demonstrating strong alignment with human judgments. |
>
> \
> \
> In addition to the major concerns above, reviewers also raised several minor points. We addressed these with targeted clarifications and revisions, summarized below.
>
>
> | Minor Concerns | Reviewers | Our Responses and Revision |
> |---|---|---|
> | **Confusion about evaluation pipeline and when judges are applied** | UA7N, rYaq, z1ru, XHKK | Introduced a **new Section 4.3 (“Judge Framework”)** clarifying the four-stage pipeline (refusal → attack evaluation → syntax → runtime) and emphasizing that syntax and runtime judges are only invoked for artifacts labeled harmful by the attack evaluation judge. |
> | **Positioning vs. prior executable-aware work (RedCode)** | UA7N, XHKK | Updated the **Introduction, Related Work, and Section 7** to (i) properly cite RedCode as earlier executable-aware work on code agents, (ii) tone down any “first” wording, and (iii) frame “agents more vulnerable than base LLMs” as consistent with RedCode, while clarifying how CAJ extends the space via multi-level workspace regimes. |
> | **Docker sandbox safety / executability infrastructure clarity** | rYaq | Clarified Docker sandbox and executability judge setup in **Section 4.2**, emphasizing that the executability judge is a non-mutating micro-agent operating on an isolated workspace inside Docker. |
> | **Language diversity / polyglot realism** | rYaq | Clarified that CAJ-Bench includes non-Python languages and that agents are free to choose languages as needed; added details and examples in **Appendix A.3** (CodeAgentJail-Bench Details). |
> | **Problem Definition section too long / overly formal** | z1ru | Shortened and streamlined the **Problem Definition** section to retain only the notation and assumptions used later in the paper. |
> | **Extending benchmark toward attacks/defenses** | UA7N | Expanded the **Conclusion** to explicitly discuss how CAJ-Bench and the Judge Framework can be extended to more sophisticated attacks and execution-aware defenses. |

---

> > ### Author Response · Authors · 2025-12-03
> > **Author Final Remarks (3/3)**
> >
> > ## IV. Revised Manuscript
> > ---
> > We include this section to help the Area Chair follow the revisions in the manuscript sequentially. We hope this mapping makes it easier to track the changes and provides a smoother review experience.
> >
> > Consistent with our rebuttal commitments, we have incorporated all corresponding updates into the revised manuscript (edits highlighted in blue). Below, we map the major revisions to specific sections/appendices.
> >
> >
> > * **Benchmark updates and CAJ-M quality validation**
> >
> >     * **Section 3 (CodeAgentJail Benchmark):** Added CAJ-M construction details and strengthened dataset quality assurance using VirusTotal Code Insight, CodeQL, and manual inspection (last paragraph of Section 3).
> >
> > * **Judge framework clarification (pipeline semantics) + plug-and-play reuse**
> >
> >     * **Section 4.1 (Robustness Judges):** Clarified refusal and attack-evaluation judging; added human alignment results for judge reliability.
> >     * **Section 4.2 (Executability Judges):** Clarified the non-mutating micro-agent design and runtime evaluation protocol.
> >     * **Section 4.3 (Judge pipeline and reuse):** Added an end-to-end description of the four-stage pipeline (refusal → attack evaluation → syntax → runtime), explicitly stating that syntax/runtime judges are only invoked for artifacts already labeled harmful, and noting the modular implementation (robustness_judge, exec_judge) enabling plug-and-play reuse.
> >
> > * **Human alignment of robustness judges**
> >
> >     * **Section 4.1:** Added author-led human verification of the Refusal and Attack Evaluation Judges and reported alignment results.
> >
> > * **Ablations supporting agent-agnosticity**
> >
> >     * **Section 7 (“Extending to Different Code Agents”):** Added an ablation extending CAJ-0 to additional agent frameworks (SWE-Agent and a Codex-style agent) to show that CAJ-Bench trends remain consistent across scaffolds (agent-agnostic).
> >
> > * **Positioning vs. related work (RedCode; Power 2025)**
> >
> >     * **Introduction:** Removed/toned down “first” wording and reframed *“agents more vulnerable than base LLMs”* as consistent with RedCode, while emphasizing our distinct contribution of multi-level workspace regimes and deployable-harm evaluation.
> >     * **Related Work:** Expanded comparison with RedCode and clarified how CAJ-Bench complements prior executable-aware evaluations.
> >     * **Section 6.3:** Added explicit discussion that “added code context helps jailbreaking” is consistent with concurrent work Power (2025) (“Drop the Guardrails”).
> >
> > * **Base-LLM vs agentic-wrapper analysis**
> >
> >     * **Section 7:** Clarified and contextualized the base-LLM vs agentic-wrapper comparison and aligned its interpretation with prior work.
> >
> > * **Docker sandbox / execution details**
> >
> >     * **Section 4.2:** Added clarifications on Docker isolation and the executability judge configuration (non-mutating micro-agent on an isolated workspace).
> >
> > * **Language diversity and benchmark details**
> >
> >     * **Appendix A.3 (CodeAgentJail-Bench Details):** Clarified that CAJ includes non-Python samples (e.g., JavaScript/C) and that agents are free to choose the implementation language where appropriate.
> >
> > * **Concise Problem Definition**
> >
> >     * **Problem Definition section:** Shortened and streamlined the formalism to retain only assumptions and notation used later in the paper.
> >
> > * **Extensions to attacks and defenses**
> >
> >     * **Conclusion:** Expanded discussion on how CAJ-Bench and the Judge Framework can be extended to more sophisticated attacks and execution-aware defenses, including refusal persistence and pre-execution safety gating.
> >
> > ---
> > ### We deeply appreciate the expertise and time of the Area Chair and reviewers.

---

### Note · Program_Chairs · 2026-01-17
**Submission Desk Rejected by Program Chairs**

The following references in this submission do not refer to real documents and/or have major errors in bibliographic information:

 Xin Wang, Yue Zhang, Shuyang Guo, Yingfei Yang, Zhong Liu, Zihan Zhu, Zixuan Zhou, Chong Zhang, Wenxuan Qian, and Pengfei Yin. Openhands: Making llms practical for hands-on code development. arXiv preprint arXiv:2212.10481, 2022.